# A study on limited pre-sale strategy with consideration of consumer regret

**Yong-chang Jiang⊙, Yue-yu Qi⊙*⊙**

School of Management, Harbin University of Commerce, Harbin, Heilongjiang, China

⊙ These authors contributed equally to this work.
* qyy@s.hrbcu.edu.cn

## Abstract

The effect of regret on consumers' purchasing behavior is more and more obvious. The limited pre-sale can make retailers with limited production capacity allocate two periods of stock effectively and increase their income. This paper considers the heterogeneous consumers with regret behavior in the market and constructs a model to study the retailer's optimal limited pre-sale strategy. The results show that the high price regret sensitivity negatively affects the higher price of the products in the pre-sale strategy, while the out-of-stock regret sensitivity negatively affects the retailer's profit When the production capacity is relatively low, the proportion of rational consumers is large and the high price regret sensitivity coefficient is small, the retailer should pre-sell at a limited discount and the lowest valuation, and the highest valuation is on sale, otherwise, it should be sold at a price slightly lower than the highest valuation, but when the capacity is very sufficient, the sensitive coefficient of stock-out regret is small and the proportion of rational consumers is small, the retailer should pre-sell at an unlimited premium, and a price slightly lower than the highest valuation of the pre-sale, the lowest valuation of the sale, or should be pre-sold at the highest valuation.

## Introduction

Pre-sale mode is a kind of sales mode in which the seller allows the buyer to buy the goods or services at a certain time before they are consumed, and the buyer has to wait for a certain period time before he can consume the goods or services. For consumers, buying in advance not only allows them to get discounts or lock-in goods ahead of time to avoid the risk of out-of-stock but also because retailers often promise to pay first and deliver first, choosing to pre-order can get goods or services earlier. For retailers, the advance sale method can help them better forecast the demand for goods [1], so that they can stock up in the spot period to effectively reduce the inventory risk [2]: when the consumer demand is uncertain and the preference is not clear, pre-sale data can mitigate Information asym-metry risks between retailers and consumers [3,4], and can be used to provide volume and profit through price discrimination [5]. With the development and popularization of online shopping, advance sale is more and more popular with retailers. Therefore, when to adopt what kind of pre-sale strategy and how to price has become a limited capacity of retailers to solve the problem.

**Data Availability Statement:** The dataset is available on Figshare: https://doi.org/10.6084/m9.Figshare.22656235.v1.

**Funding:** Project supported by the Heilongjiang Provincial Social Science Foundation Project(No. 18TQD365). The funders had no role in study

design, data collection and analysis, decision to publish, or preparation of the manuscript.

**Competing interests:** The authors have declared that no competing interests exist.

Pre-sale strategy can be divided into discount pre-sale and premium pre-sale ac-cording to the relationship between the pre-sale period and the spot price. Premium pre-sale refers to the pre-sale when the price of the product is higher than the current price. Because for a limited number of service products, in order to avoid the risk of spot short-age, consumers are often willing to pay a premium over the spot price in advance to lock-in products. For example, loyal users of some electronic products are willing to pay a premium for access to the product in the first place. On the contrary, retailers will lower prices during the pre-sale stage to encourage consumers to buy products during the pre-sale stage. For the study object of this paper, that is, the limited capacity or service ca-pacity and can not be improved in a short period of time, under certain conditions to limit the pre-sale period of sales can make retailers increase revenue [6], that is, limited pre-sale. Under the constraint of production capacity, restricting the pre-sale volume has become a popular operation mode under the pre-sale mode.

The intertemporal price discrimination of retailers may increase the income, but it will affect the consumer's purchasing decision. Therefore, intertemporal price combina-tion should consider the consumer's behavior. From the perspective of consumers, on the one hand, the choice of pre-sale purchase can ensure the availability of products, but once the nor-mal sale phase can buy goods, consumers will pay more money for pre-sale period regret; On the other hand, if consumers choose to wait until the normal sale stage to buy, then will face the possibility of out-of-stock, once out of stock, consumers will regret their waiting behavior, bemoan the loss of utility caused by waiting. Because of the change in utility caused by regret, strategic consumers begin by calculating the expected utility of their purchases at various stages in order to choose to buy at the time of their Utility maximization problem.

In this paper, we will study the limited pre-sale strategy of retailers with limited ca-pacity or limited service capacity, and focus on the optimal decision-making under dif-ferent capacity conditions.

To sum up, this paper has the following innovations compared with the previous lit-erature: (1) Considering the impact of consumer regret on heterogeneous consumer buying behavior, this paper explores how retailers with limited capacity should choose pre-sale strategy. (2) By comparing the feasible conditions of no-pre-sale, discount-pre-sale and premium-pre-sale, the optimal pre-sale strategy and the corresponding pricing decision are given.

The rest of this article is organized as follows. The second part is literature review. The third part describes the specific parameters and assumptions of the model. In the fourth part, the profit functions of non-pre-sale, premium pre-sale and discount pre-sale are constructed. The fifth part compares the advantages and disadvantages of the three strategies, and obtains the retailer's strategies under different conditions. The sixth part summarizes the research content of this paper, and puts forward the deficiency and the research direction in the future.

## Literature review

The literature related to this study mainly deals with two aspects: pre-sale strategy and con-sumer regret.

### Pre-sale strategy

Early research on pre-sale models focused on the service industry, commonly pre-sales of vari-ous types of tickets. Jinhong X and Steven M showed that offering pre-sales is beneficial for firms, especially when consumers are uncertain about the value of the product [7]. Yu M et al. demonstrated that pre-sale may be advantageous when there is a large number of consumers and a high degree of individual valuation diversification [8]. Li C et al explored the pre-sale

decision in a pre-sale model when both short-sighted and strategic consumers are present [9]. Then Li H and Qi ES comparatively studied pre-selling and non-pre-selling strategies so as to give the conditions for retailers to implement pre-selling strategies [10]. However, there is little literature on limited pre-sale. After that, Chen J and Bell PC studied the demand uncertainty of fixed capacity and limited product choice for fugitive products [11]. Zhang M et al explored the limited pre-sale model for new products on flash sale platforms [12]. Ji GJ and so on introduced the reference price effect in the pre-sale model, thus obtained the retailer's optimal pre-sale decision [13], and then continued to study the ref-erence price effect on the limited pre-sale strategy [14].

## Consumer regret

Expected regret reflects a combination of regret, depression, disappointment, remorse, and self-blame that will be experienced by not buying the limited product [15–17], coupled with the higher perceived value of the product that comes with a production-based limited [18] and the stronger motivation for consumers to own the product [19]. Thus, compared to sales-based limits, production-based limits are more likely to induce strong feelings of expected ela-tion and expected regret. In the area of consumer regret research, Zen Z and Zheng Y consid-ered the high price that consumers would face if they bought immediately after the full-price period and the stock-out that they would face if they waited until the liquidation period [20]. Then, Adida E and ÖZER Ö, on the basis of Zen Z and Zheng Y, considered the effect of antic-ipatory regret behavior on firms' pricing strategy choice in the competitive market environ-ment, and emphasized the advantage of price reduction strategy [21]. Chen FJ and Guan ZZ studied the impact of expected regret on return strategies when consumers' valuations are uncertain [22]. Zhou JH et al analyzed the impact of high price regret and out-of-stock regret on the pricing strategy of enterprises [23]. Shi BL and other considerations when consumers pre-purchase regret behavior does not provide returns, returns no longer on sale and return re-sale of pre-sale strategy [24].

Based on the above literature, we know that regret emotion does affect the retailer's limited pre-sale strategy, so it is meaningful to study the limited pre-sale strategy under consumer regret.

## Model description and assumptions

This paper considers a product market consisting of a monopoly retailer and a consumer. Monopolistic retailers adopt a two-stage sales model in the process of selling a single product and develop a pre-sale strategy based on the principle of profit maximization, the correspond-ing product consumers from their own utility to make a favorable purchase decision.

Assuming that the capacity of retailers is limited and can not be significantly increased in a short period of time, its current capacity is defined as $C \in (0,1)$, and it has the ability to imple-ment pre-sale. The retailer adopts a price commitment and two-stage pre-sale strategy, and announces the pre-sale price and the current sale price of the product during the pre-sale stage, and informs the consumer whether the pre-sale quantity is limited to $K(K \leq C)$. The first stage is pre-sale stage, the product price is $P_1$, The second stage is the selling stage, the product price is $P_2$. The aim of limited pre-sales is to rationalise the distribution of product inventories and to improve profits by balancing supply and demand to minimise consumer Surplus value [14].

Suppose that all consumers in the market are 1, and arrive in two periods, and obey uni-form distribution [7], that is, the number of customers arriving in the pre-sale stage and the present-sale stage is the same, is 1/2. The customers who arrive at the pre-sale stage know the pre-sale information in advance and can buy or wait until the pre-sale stage, but the customers

who arrive at the pre-sale stage can only choose to buy or leave directly at the pre-sale stage. Suppose that the consumer's valuation of the product follows a two-point distribution [25], that is, $v \in \{V_L, V_H\}$, and the probability of getting a high valuation $V_H$ is $q$, the probability of getting a low valuation $V_L$ is $1-q$ [7,25]. According to the research of Ji Guo-jun [13,14], supposing that $1/2 \leq V_L/V_H < 1$, $1/2 \leq q < V_H/V_L$, it can be understood that: the probability $q$ of high valuation of consumers is too large, retailers should ignore the demand of low-valuation consumers and sell at high prices; Conversely, the probability of undervaluation $1-q$ is too high, and retailers sell at rock-bottom prices, and it is uneconomic. For the sake of calculation, let $q = 1/2$, then $EV = (V_H+V_L)/2$. For the retailer, when the pre-sale price $P_1 < V_L$ or the selling price $P_2 < V_L$, the profit of a single product is too small and uneconomic. And when $P_1 > V_H$ or $P_2 > V_H$, no consumer purchases, so none of the above would be optimal pricing strategy. Therefore, $P_1, P_2 \in [V_L, V_H]$.

According to Xu and Duan [26], it is assumed that there are both rational and emotional consumers in the market. Among them, the proportion of rational consumers is $a$, which only considers the economic utility obtained from the pre-sale and the current sale stage, and there is no regret behavior; the proportion of emotional consumers is $\tilde{a} = 1 - a$, which considers not only the economic utility when purchasing, but also the economic utility, we also need to consider the psychological effects of regretful behavior [22]. To facilitate research, each consumer can only buy one unit of the product. According to the research of the literature [21], the perceptual consumer's regret behavior is divided into two kinds: one is the high price regret, that is, the loss after the consumer buys the product at a high price, which is expressed by $\beta$ as the sensitive coefficient of the high price regret; the other is the loss of out-of-stock regret, that is, the loss caused by retailers' out-of-stock when consumers buy, is expressed by $\gamma$ as the sensitive coefficient of out-of-stock regret. For consumers, getting the product is more important than buying it at a low price [20,21], so assume $\gamma > \beta > 0$.

The literature related to this study mainly deals with two aspects: pre-sale strategy and consumer regret.

## Model construction and analysis

It constructs the model under the three strategies of no-pre-sale, discount-pre-sale and premium-pre-sale which the retailer may choose, and find out the optimal pricing and limited quantity strategy of the retailer under each strategy, the optimal decision of retail-ers is obtained by comparison.

### No pre-sale strategy

If the retailer adopts the no-pre-sale strategy, the price is equal and the quantity is unlimited [14], so the consumers have no regret loss, then the purchasing utility of both kinds of consumers is equal $U = v-P$ [26]. So retailers can use two pricing strategies: strategy 1: $P = V_L$; strategy 2: $P = V_H$.

When the retailer adopts strategy 1 and all consumers buy, the profit function is $\pi_1 = V_L C$. When the retailer adopts Strategy 2, only high-valued consumers will buy, then the profit function is $\pi_2 = V_H min\{C, 1/2\}$.

When $C \leq 1/2$, Strategy 1 sells all products short at low price $V_L$, Strategy 2 sells all products short at high price $V_H$, and when $C > 1/2$, Strategy 1 sells all products short at low price $V_L$, strategy 2 sells 1/2 of the product at a high price $V_H$. By comparing the profits of the two strategies, we obtain theorem 1.

Theorem 1. When a retailer adopts a no-pre-sale strategy, Strategy 1 is better than strategy 2 when $C > \frac{V_H}{2V_L}$, and Strategy 2 is better than strategy 1 when $C \leq \frac{V_H}{2V_L}$.

Proof of Theorem 1. The proof process is shown in Appendix A in S1 Appendix.

Theorem 1 shows that consumers' regret has no effect on the price when the retailer does not pre-sell, and there is no need to limit the quantity. In addition, when the capacity of the retailer is relatively small, the retailer can ignore the low-valued consumers, only target the high-valued consumers, and sell the products at high prices to make up for the loss of profits caused by the remaining products; When production capacity is relatively large, it should be targeted at all consumers, all products sold at low prices to enhance profits.

## Discount pre-sale strategy

When the retailer adopts the discount pre-sale strategy, $P_1 < P_2$. For all consumers, only when the utility of purchasing is not less than 0 will they choose to buy products, and for retailers, the optimal pricing strategy is to make the utility of consumers exactly 0, this is when the retailer gets the most profit [14].

Firstly, this paper analyzes the relationship between the production capacity of retailers and the demand of consumers. There are three situations: (1) if the production capacity is lower than the demand at the present sale stage, the demand at the pre-sale stage is 0, it should not be pre-sold. (2) if the production capacity is higher than the demand in the pre-sale stage but can not meet all the demand in the pre-sale stage, the limited pre-sale stage can be chosen, and the optimal limit in the pre-sale stage is the difference between the production capacity and the demand in the pre-sale stage. (3) when the production capacity can meet all consumer demand, can be limited or unlimited pre-sale. Therefore, $\theta(0 < \theta \leq 1)$ is used to express the probability that the consumer can obtain the product in the pre-sale stage.

Then analyze the two types of consumers in the two-stage purchase behavior. For the rational consumer, the purchase utility $U_1^a = \theta(v - P_1)$ at the pre-sale stage, and the purchase utility $U_2^a = v - P_2$ at the present sale stage. The optimal two-stage pricing obtained from $U_1^a = 0$ and $U_2^a = 0$ is $P_1^* = v$ and $P_2^* = v$. For emotional consumers, the purchase utility $U_1^{1-a} = \theta(v - P_1) - \gamma(1 - \theta)max\{0, v - P_2\}$ at the pre-sale stage, and the purchase utility $U_2^{1-a} = v - P_2 - \beta\theta(P_2 - P_1)$ at the pre-sale stage. Among them, $\gamma(1-\theta)max\{0, v-P_2\}$ is the utility loss caused by out-of-stock regret, $\beta\theta(P_2 - P_1)$ is the utility loss caused by high-price regret. From $U_1^{1-a} = 0$ and $U_2^{1-a} = 0$, the optimal pricing of two stages is $P_1^* = v - \frac{\gamma(1-\theta)max\{0, v-P_2\}}{\theta}$ and $P_2^* = \frac{v + \beta\theta P_1}{1 + \beta\theta}$.

Because $P_1 < P_2$ and $P_1, P_2 \in [V_L, V_H]$, the retailer can adopt the following pricing strategy:

Strategy 3: $P_{1-3}^* = V_L$, $P_{2-3}^* = V_H$; strategy 4: $P_{1-4}^* = V_L$, $P_{2-4}^* = \frac{V_H + \beta\theta V_L}{1 + \beta\theta}$.

When retailers adopt strategy 3, all consumers with low valuations buy only at the pre-sale stage. High-valued rational consumers in the pre-sale stage of the purchase utility is greater than 0, in the current sale stage of the purchase utility is 0, then it can be in the pre-sale stage or wait until the current sale stage to buy. High-valued emotional consumers in the pre-sale stage of utility greater than 0, in the current sale stage of utility less than 0, so it will only buy in the pre-sale stage.

At the selling stage, all the low-valued consumers leave, the high-valued rational consumers do not buy, the high-valued emotional consumers leave.

Through the above analysis of consumers in the two-stage purchase can be obtained theorem 2.

Theorem 2. When the retailer adopts strategy 3, (1) if $C \leq a/2$, the retailer does not pre-sell. (2) If $a/2 < C \leq 1/2$, the retailer sells in advance in a limited quantity, and the optimal amount of presale is $K_3^* = C - a/2$. (3) If $C > 1/2$, the retailer sells in advance in a limited quantity, and the optimal advance quantity $K_3^* = (2 - a)/4$.

Proof of Theorem 2. The proof process is shown in Appendix A in S1 Appendix.

When the retailer adopts strategy 4, the low-valued consumer arriving at the pre-sale stage will only buy at the pre-sale stage. And the high-valued rational consumer because the pre-sale stage and the current sale stage of the purchase utility is greater than 0, so can buy in the pre-sale stage, or wait until the current sale stage to buy. High-valued emotional consumers in the pre-sale stage of the utility is greater than 0, in the current sale stage of the utility is equal to 0, so can also buy in the pre-sale stage, or wait until the current sale stage to buy.

At the present sale stage, all low-valued consumers leave, choose not to buy, and all high-valued consumers buy.

Through the analysis of consumer's purchasing behavior in two stages, we can get theorem 3.

Theorem 3. When the retailer adopts strategy 4, (1) if $C \leq 1/2$, the retailer does not pre-sell. (2) If $1/2 < C \leq 3/4$, the retailer sells in advance in a limited quantity, and the optimal amount of presale is $K_4^* = C - 1/2$. (3) If $C > 3/4$, the retailer sells in advance in a limited quantity, and the optimal advance quantity is $K_4^* = 1/4$.

Proof of Theorem 3. The proof process is shown in Appendix A in S1 Appendix.

Combining theorem 2 and theorem 3, we can get theorem 4.

Theorem 4. When the retailer adopts the discount pre-sale strategy, (1) if $C \leq 1/2$, the retailer does not pre-sell. (2) If $\frac{1}{2} < C \leq \frac{3}{4}$ and $a > a_1$, Strategy 3 is better. The retailer sells in advance in a limited quantity, and the optimal amount of presale is $K_3^* = C - a/2$. (3) If $\frac{1}{2} < C \leq \frac{3}{4}$ and $a \leq a_1$, Strategy 4 is better. The retailer sells in advance in a limited quantity, and the optimal advance quantity is $K_4^* = C - 1/2$. (4) If $C > \frac{3}{4}$ and $a > a_2$, Strategy 3 is better. The retailer sells in advance in a limited quantity, and the optimal amount of presale is $K_3^* = (2 - a)/4$. (5) If $C > \frac{3}{4}$ and $a \leq a_2$, Strategy 4 is better. The retailer sells in advance in a limited quantity, and the optimal advance quantity is $K_4^* = 1/4$.

Among them, $a_1 = \frac{2(V_H - 2V_L + 2CV_L + 2\beta V_L - 8C\beta V_L + 8C^2 \beta V_L)}{(1 - 2\beta + 4C\beta)(2V_H - V_L)}$, $a_2 = \frac{2V_H - V_L + \beta V_L}{(1 + \beta)(2V_H - V_L)}$.

Proof of Theorem 4. The proof process is shown in Appendix A in S1 Appendix.

Theorem 4 shows that, first of all, when retailers use discount pre-sale, they should limit pre-sale in order to attract more consumers (especially those with high valuations) to buy at the present sale stage because the present sale price is higher. So when the capacity is less than the demand at the selling stage, the retailer should sell at the highest price at the selling stage. Secondly, when the production capacity can meet the needs of the current stage and part of the pre-sale stage of demand, do not produce surplus, retailers should take limited pre-sale, can effectively allocate inventory, increase revenue by luring more consumers to the on-sale stage. In limited pre-sale, some high-valued rational consumers who arrive at the pre-sale stage are forced to wait until the pre-sale stage to purchase at a higher price because they can afford a higher spot price. Third, the rational consumer with high valuation may choose to buy at the pre-sale stage or the on-sale stage, while the emotional consumer with high valuation under strategy 3 will only buy at the pre-sale stage, it proves that consumers' regret does affect their purchasing behavior.

Therefore, regardless of whether the capacity of the retailer is more substantial or sufficient, as long as the proportion of rational consumers is enough, the retailer can ignore part of the emotional consumers and adopt strategy 3, the emotional consumer group can not be ignored, should use strategy 4, with a lower spot price to attract its purchase at the present sale stage. Finally, when the product can meet all the needs of consumers, the product has surplus, although unlimited can attract more consumers to buy, but because some consumers will switch to the pre-sale stage to buy, in order to maximize the profit, the pre-sale should be limited, and the pre-sale volume should be set as part of the demand only in the pre-sale stage. In a word, the retailer should limit the quantity while taking the discount pre-sale.

## Premium pre-sale strategy

When the retailer adopts the premium pre-sale strategy, $P_1 > P_2$. Firstly, the relationship between the capacity of retailers and the demand of consumers is analyzed. There are three situations: if the capacity is lower than the demand in the pre-sale stage, the demand in the present sale stage is 0, which can be equivalent to no pre-sale; If the production capacity is higher than the pre-sale stage demand but can not meet all the needs of the pre-sale stage, unlimited pre-sale, because the pre-sale price is high, to allow more consumers to buy in the pre-sale stage; When the production capacity to meet all consumer demand, unlimited pre-sale. Therefore, $\mu(0 < \mu \leq 1)$ is used to express the probability that the consumer can get the product at the present sale stage [14].

Then analyze the two types of consumers in the two-stage purchase behavior. For the rational consumer, the purchase utility $U_1^a = v - P_1$ in the pre-sale stage, the purchase utility $U_2^a = \mu(v - P_2)$ in the present sale stage, the optimal two-stage pricing obtained from $U_1^a = 0$ and $U_2^a = 0$ is $P_1^* = v$ and $P_2^* = v$. For emotional consumers, the purchasing utility $U_1^{1-a} = v - P_1 - \beta\mu(P_1 - P_2)$ at the pre-sale stage, and $U_2^{1-a} = \mu(v - P_2) - \gamma(1 - \mu)max\{0, v - P_1\}$ at the present sale stage. Among them, $\gamma(1-\mu)max\{0, v-P_1\}$ is the utility loss caused by out-of-stock regret, $\beta\mu(P_1 - P_2)$ is the utility loss caused by high-price regret. From $U_1^{1-a} = 0$ and $U_2^{1-a} = 0$, the two-stage optimal pricing is $P_1^* = \frac{v + \beta\mu P_2}{1 + \beta\mu}$ and $P_2^* = v + \frac{\gamma(-1+\mu)max\{0, v - P_1\}}{\mu}$.

Because $P_1 < P_2$, $v \in \{V_L, V_H\}$ and $P_1, P_2 \in [V_L, V_H]$, the retailer can adopt the following pricing strategy:

Strategy 5: $P_{1-5}^* = V_H$, $P_{2-5}^* = V_L$. Strategy 6: $P_{1-6}^* = \frac{V_H + \beta\mu V_L}{1 + \beta\mu}$, $P_{2-6}^* = V_L$.

When retailers adopt strategy 5, in the pre-sale stage, all consumers with low valuations wait until the current sale stage to buy. The utility of the highly valued rational consumer is zero at the pre-sale stage and greater than zero at the present sale stage. To make the retailer more profitable, suppose they buy the product at the pre-sale stage. Highly valued emotional consumers have a utility of less than 0 at the pre-sale stage and 0 at the current sale stage, so they will wait until the current sale stage to buy. At the present sale stage, all consumers buy.

Through the analysis of the above two-stage purchase of consumers, it can be seen that when the retailer adopts strategy 5, when $C \leq a/4$, the retailer does not pre-sell, when $C > a/4$, the retailer does not pre-sell.

When retailers adopt strategy 6, all consumers with low valuations at the pre-sale stage will only wait until the pre-sale stage to buy. A rational consumer with a high valuation has a purchase utility greater than 0 at both the pre-sale stage and the current sale stage, so it can buy at the pre-sale stage or wait until the current sale stage. The utility of the highly valued perceptual consumer at the pre-sale stage is equal to 0, and the utility at the current sale stage is $U_2^{1-a} = \frac{\mu(1 + \beta(\gamma(-1+\mu)+\mu))(V_H - V_L)}{1 + \beta\mu}$. If $U_2^{1-a} > 0$, then $\gamma \leq \frac{-1 - \beta\mu}{\beta(-1+\mu)}$.

Discuss in two situations:

Situation 1: $\gamma \leq \frac{-1 - \beta\mu}{\beta(-1+\mu)}$.

At this point, high-valued emotional consumers in the current sale stage utility greater than 0, so it can be bought in the pre-sale stage, but also can wait until the current sale stage to buy. At the present sale stage, all consumers buy.

Through the above two-stage analysis of consumer purchases, when the retailer adopts strategy 6, if $C \leq 1/4$, the retailer does not pre-sell. If $C > 1/4$, the retailer is not limited to pre-sale. Because of the large demand, suppose that the high-valued consumers who arrive at the pre-sale stage buy at the pre-sale stage, and $\mu = \frac{4}{3}C - \frac{1}{3}$.

According to the profit comparison between strategy 5 and Strategy 6, the optimal decision on premium pre-sale is obtained, which is theorem 5.

Theorem 5. If $\gamma \leq \frac{-1-\beta\mu}{\beta(-1+\mu)}$ and the retailer sells in advance at a premium, (1) when $C \leq 1/4$, the retailer does not sell in advance. (2) when $C > 1/4$ and $a > a_3$, strategy 5 is better. (3) when $C > 1/4$ and $a \leq a_3$, strategy 6 is better. And both strategies do not use limited pre-sale.

Among them, $a_3 = \frac{3}{3-\beta+4C\beta}$.

Proof of Theorem 5. The proof process is shown in Appendix A in S1 Appendix.

Theorem 5 shows that when the retailer uses premium pre-selling and the emotional consumers are less sensitive to out-of-stock, the prices of the two strategies are the lowest price at the present selling stage, and all consumers will buy, at this point, no matter how much capacity the retailer has, there is no surplus. Second, when the capacity is low, retailers should not pre-sale, and when the capacity is high, the proportion of rational consumers, strategy 5 will be better than strategy 6. This is because the price of strategy 5 ignores the influence of some emotional consumers' purchasing behavior, and chooses to pre-sell at the highest price, thus improving the retailer's income. On the other hand, emotional consumers should not be ignored to sell at a lower price.

Situation 2: $\gamma > \frac{-1-\beta\mu}{\beta(-1+\mu)}$.

At this point, high-valuation emotional consumers in the current sale stage utility less than 0, so it is only in the pre-sale stage to buy. At the selling stage, all consumers except high-valued emotional consumers buy.

Through the above two-stage analysis of consumer purchase, when the retailer adopts strategy 6, when $C \leq 1/4$, the retailer does not pre-sell.

When $1/4 < C \leq (3+a)/4$, the retailer does not limit the pre-sale. Because of the high demand, suppose that the highly valued consumer arriving at the pre-sale stage buys at the pre-sale stage, and $\mu = (4C-1)/(2+a)$.

When $C > (3+a)/4$, the retailer is not limited to pre-sale. Because of the high demand, let's assume that the highly valued consumers who arrive at the pre-sale stage buy at the pre-sale stage, and $\mu = 1$.

According to the profit comparison between strategy 5 and Strategy 6, the optimal decision on premium pre-sale is obtained, which is theorem 6.

Theorem 6. When $\gamma > \frac{-1-\beta\mu}{\beta(-1+\mu)}$ and the retailer uses premium pre-sale, (1) if $C \leq 1/4$, the retailer does not pre-sell. (2) if $1/4 < C \leq (3+a)/4$ and $\beta > \beta_1$, strategy 5 is better. (3) if $1/4 < C \leq (3+a)/4$ and $\beta \leq \beta_1$, strategy 6 is better. (4) if $C > (3+a)/4$ and $\beta > \beta_2$, strategy 5 is better. (5) if $C > (3+a)/4$ and $\beta \leq \beta_2$, strategy 6 is better. And both strategies do not use limited pre-sale.

Among them, $\beta_1 = \frac{2-a-a^2}{a(-1+4C)}$, $\beta_2 = \frac{-(-1+a)V_H+2(1+a-2C)V_L}{aV_H+(-3-2a+4C)V_L}$.

Proof of Theorem 6. The proof process is shown in Appendix A in S1 Appendix.

Theorem 6 shows that when retailers use premium pre-selling and perceptual consumers are more sensitive to out-of-stock, consumers under strategy 5 will buy, and there will be no surplus, and strategy 6 under the current sale stage to reach the high-valued emotional consumer will not buy the product, the product may be surplus. Second, when capacity is low, retailers should not pre-sell, and when capacity is more or more sufficient, emotional consumers are more sensitive to high price regret, strategy 5 will be better than strategy 6. This is because the more sensitive the emotional consumers are to high price regrets, the more they may not buy the products at the current selling stage, resulting in a loss of profit for the retailers. Therefore, some needs of the emotional consumers should be ignored and the products should be pre-sold at the highest price, to make up for the resulting loss of profits.

Combining theorem 5 and theorem 6, it can be concluded that when retailers use premium pre-sale, the pre-sale price is higher than the current sale price, and it is an effective way to

increase profits to sell as many products as possible in the pre-sale stage, so it is not appropriate to limit the pre-sale.

## The profit function of three pre-sale strategies

Combined with Theorem 1–6, the profit functions of three pre-sale strategies shown in Table 1 are obtained.

## Optimal pre-sale decision of a retailer

By comparing the advantages and disadvantages of no-pre-sale, discount-pre-sale and premium-pre-sale strategies, the optimal pricing decisions for retailers under different circumstances are determined.

Combining theorem 1,4,5 and 6, we get theorem 7.

Theorem 7. When $C \leq 1/2$, no pre-sale is better than premium pre-sale and discount pre-sale.

Proof of Theorem 7. The proof process is shown in Appendix A in S1 Appendix.

Theorem 7 shows that when the capacity of the retailer is low (less than 1/2), it is not suitable to sell in advance, but should sell at the highest price.

Then, a numerical example is given to illustrate the relationship between optimal decision-making of retailers and relevant parameters. Theorem 7 states that when the production capacity is less than 1/2, no pre-sale is used, so we mainly analyze the relationship between the profit of retailers under different strategies when $C>1/2$, and get the corresponding optimal decision. Set the following parameters according to the actual situation: $V_L = 6$, $V_H = 10$.

Let $C = [0.51, 0.6, 0.7, 0.8, 0.9, 0.99]$, $a = [0.3, 0.5, 0.7]$, $\beta = 1$. The results of the calculation are shown in Table 2.

**Table 1. The profit functions of three pre-sale strategies.**

| strategies | | C | profit |
|---|---|---|---|
| **No pre-sale strategy** | $\pi^*$ | $C \leq 1/2$ | $V_H C$ |
| | | $1/2 < C \leq \frac{V_H}{2V_L}$ | $\frac{V_H}{2}$ |
| | | $C > \frac{V_H}{2V_L}$ | $V_L C$ |
| **Strategy 3** | $\pi_3^*$ | $C \leq \frac{a}{2}$ | $V_H C$ |
| | | $\frac{a}{2} < C \leq \frac{1}{2}$ | $V_L\left(C - \frac{a}{2}\right) + V_H \frac{a}{2}$ |
| | | $C > \frac{1}{2}$ | $V_L \frac{2-a}{4} + V_H \frac{a}{2}$ |
| **Strategy 4** | $\pi_4^*$ | $C \leq \frac{1}{2}$ | $V_H C$ |
| | | $\frac{1}{2} < C \leq \frac{3}{4}$ | $V_L\left(C - \frac{1}{2}\right) + \frac{V_H + \beta(4C-2)V_L}{1+\beta(4C-2)}\frac{1}{2}$ |
| | | $C > \frac{3}{4}$ | $V_L \frac{1}{4} + \frac{V_H + \beta V_L}{1+\beta}\frac{1}{2}$ |
| **Strategy 5** | $\pi_5^*$ | $C \leq \frac{a}{4}$ | $V_H C$ |
| | | $C > \frac{a}{4}$ | $V_L\left(C - \frac{a}{4}\right) + V_H \frac{a}{4}$ |
| **Strategy 6** | If $\gamma \leq \frac{-1-\beta\mu}{\beta(-1+\mu)}$, $\pi_6^*$ | $C \leq \frac{1}{4}$ | $V_H C$ |
| | | $C > \frac{1}{4}$ | $\frac{3V_H + (4C-1)\beta V_L}{3+(4C-1)\beta}\frac{1}{4} + V_L\left(C - \frac{1}{4}\right)$ |
| | If $\gamma > \frac{-1-\beta\mu}{\beta(-1+\mu)}$, $\pi_6^{*\prime}$ | $C \leq \frac{1}{4}$ | $V_H C$ |
| | | $\frac{1}{4} < C \leq \frac{3+a}{4}$ | $\frac{(2+a)V_H + (-1+4C)\beta V_L}{2+a+(-1+4C)\beta}\frac{1}{4} + V_L\left(C - \frac{1}{4}\right)$ |
| | | $C > \frac{3+a}{4}$ | $\frac{V_H + \beta V_L}{1+\beta}\frac{1}{4} + V_L \frac{2+a}{4}$ |

https://doi.org/10.6084/m9.figshare.22566592.v1.

Table 2. When $\beta = 1$, the profits of each strategy vary with C when a value is different.

| $\alpha$ | | C | 0.51 | 0.6 | 0.7 | 0.8 | 0.9 | 0.99 |
|---|---|---|---|---|---|---|---|---|
| 0.3 | | No pre-sale strategy | 5 | 5 | 5 | 5 | 5.4 | 5.94 |
| | | Strategy 3 | 4.05 | 4.05 | 4.05 | 4.05 | 4.05 | 4.05 |
| | | Strategy 4 | 4.98 | 5.03 | 5.31 | 5.5 | 5.5 | 5.5 |
| | | Strategy 5 | 3.36 | 3.9 | 4.5 | 5.1 | 5.7 | 6.24 |
| | When $\gamma$ is smaller | Strategy 6 | 3.8 | 4.28 | 4.825 | 5.38 | 5.94 | 6.44 |
| | When $\gamma$ is larger | | 3.75 | 4.22 | 4.76 | 5.31 | 5.45 | 5.45 |
| 0.5 | | No pre-sale strategy | 5 | 5 | 5 | 5 | 5.4 | 5.94 |
| | | Strategy 3 | 4.75 | 4.75 | 4.75 | 4.75 | 4.75 | 4.75 |
| | | Strategy 4 | 4.98 | 5.03 | 5.31 | 5.5 | 5.5 | 5.5 |
| | | Strategy 5 | 3.56 | 4.1 | 4.7 | 5.3 | 5.9 | 6.44 |
| | When $\gamma$ is smaller | Strategy 6 | 3.8 | 4.28 | 4.825 | 5.38 | 5.94 | 6.44 |
| | When $\gamma$ is larger | | 3.77 | 4.24 | 4.78 | 5.33 | 5.75 | 5.75 |
| 0.7 | | No pre-sale strategy | 5 | 5 | 5 | 5 | 5.4 | 5.94 |
| | | Strategy 3 | 5.45 | 5.45 | 5.45 | 5.45 | 5.45 | 5.45 |
| | | Strategy 4 | 4.98 | 5.03 | 5.31 | 5.5 | 5.5 | 5.5 |
| | | Strategy 5 | 3.76 | 4.3 | 4.9 | 5.5 | 6.1 | 6.64 |
| | When $\gamma$ is smaller | Strategy 6 | 3.8 | 4.28 | 4.825 | 5.38 | 5.94 | 6.44 |
| | When $\gamma$ is larger | | 3.78 | 4.25 | 4.8 | 5.35 | 5.90 | 6.05 |

https://doi.org/10.6084/m9.figshare.22566610.v1.

The results in Table 2 show that: (1) the profits of all strategies except strategy 3 will increase to some extent with the increase of the retailer's capacity. (2) with the exception of no pre-sale, Strategy 4 and Strategy 6($\gamma$ is smaller), the profits of all other strategies will increase with the increase of the proportion of rational consumers. (3) comparing the profits of strategy 6 when $\gamma$ is small and $\gamma$ is large, it is found that the profits of strategy 6 will decrease correspondingly when the consumers are more sensitive to out-of-stock, and the margin will become more obvious with the increase of production capacity.

Then, the optimal decision of retailers is put forward when the consumers regret the high price more sensitively: (1) when the proportion of rational consumers is small, with the increasing of production capacity, the optimal strategy of retailers changes from not pre-selling to strategy 4, finally turned to strategy 56. This is because with the increase of production capacity, the loss of product surplus can not make up for the profit generated by the high price, so retailers should adopt the pre-sale strategy to increase sales volume and thus increase revenue. The optimal strategy is strategy 5 when the production capacity is sufficient and the consumers are more sensitive ($\gamma$ is larger) to shortage, and strategy 6 when the production capacity is sufficient and the consumers are less sensitive to shortage. This is because when consumers are more sensitive to out-of-stock, there will be high-valued emotional consumers do not buy products, then retailers should give up this part of the consumer, with high prices to make up for the loss of profits. (2) when the proportion of rational consumers is large (such as a = 0.7), the optimal strategy of retailers changes from strategy 3 to strategy 5 with the increase of production capacity. This suggests that the higher the proportion of rational consumers, the greater the benefits of price discrimination, the better the effect.

Let $C = [0.51, 0.6, 0.7, 0.8, 0.9, 0.99]$, $a = [0.3, 0.5, 0.7]$, $\beta = 0.5$. The results of the calculation are shown in Table 3.

**Table 3. When β = 0.5, the profits of each strategy vary with C when a value is different.**

| α | | C | 0.51 | 0.6 | 0.7 | 0.8 | 0.9 | 0.99 |
|---|---|---|---|---|---|---|---|---|
| 0.3 | | No pre-sale strategy | 5 | 5 | 5 | 5 | 5.4 | 5.94 |
| | | Strategy 3 | 4.05 | 4.05 | 4.05 | 4.05 | 4.05 | 4.05 |
| | | Strategy 4 | 5.02 | 5.27 | 5.63 | 5.83 | 5.83 | 5.83 |
| | | Strategy 5 | 3.36 | 3.9 | 4.5 | 5.1 | 5.7 | 6.24 |
| | When γ is smaller | Strategy 6 | 3.91 | 4.4 | 4.97 | 5.53 | 6.10 | 6.61 |
| | When γ is larger | | 3.88 | 4.37 | 4.92 | 5.48 | 5.62 | 5.62 |
| 0.5 | | No pre-sale strategy | 5 | 5 | 5 | 5 | 5.4 | 5.94 |
| | | Strategy 3 | 4.75 | 4.75 | 4.75 | 4.75 | 4.75 | 4.75 |
| | | Strategy 4 | 5.02 | 5.27 | 5.63 | 5.83 | 5.83 | 5.83 |
| | | Strategy 5 | 3.56 | 4.1 | 4.7 | 5.3 | 5.9 | 6.44 |
| | When γ is smaller | Strategy 6 | 3.91 | 4.4 | 4.97 | 5.53 | 6.10 | 6.61 |
| | When γ is larger | | 3.89 | 4.38 | 4.94 | 5.49 | 5.92 | 5.92 |
| 0.7 | | No pre-sale strategy | 5 | 5 | 5 | 5 | 5.4 | 5.94 |
| | | Strategy 3 | 5.45 | 5.45 | 5.45 | 5.45 | 5.45 | 5.45 |
| | | Strategy 4 | 5.02 | 5.27 | 5.63 | 5.83 | 5.83 | 5.83 |
| | | Strategy 5 | 3.76 | 4.3 | 4.9 | 5.5 | 6.1 | 6.64 |
| | When γ is smaller | Strategy 6 | 3.91 | 4.4 | 4.97 | 5.53 | 6.10 | 6.61 |
| | When γ is larger | | 3.90 | 4.40 | 4.95 | 5.51 | 6.08 | 6.22 |

https://doi.org/10.6084/m9.figshare.22566613.v1.

The results in Table 3 are similar to those in Table 2, and the optimal decision-making of retailers is proposed when consumers' high price regret is less sensitive: (1) when the rational consumers are small and the production capacity is sufficient, with the increase of the retailer's capacity, the retailer's optimal strategy changes from no-pre-sale to strategy 4. This is because consumers' buying behavior is affected by regret. As production capacity increases, high prices do not bring enough profit to make up for the loss caused by product surplus. (2) when the rational consumers are small and the production capacity is sufficient, if the perceptual consumers are more sensitive to the shortage, the optimal strategy of the retailer is strategy 5, and if the rational consumers are less sensitive to the shortage, the optimal strategy is strategy 6. (3) when the proportion of rational consumers is high, the optimal strategy of retailers changes according to the Order of 345 with the increase of production capacity. Whether the consumers are sensitive to out-of-stock or not, strategy 5 is optimal when the production capacity is sufficient. This is because the ideal consumers occupy a higher proportion at this time, and the buying behavior of the consumers is less affected by regret, the gains to retailers from higher prices can compensate for the loss of profits caused by emotional consumers not buying products.

Comparing the results of Tables 2 and 3, we can find that the consumer's sensitivity coefficient to high price regret will only affect the profits of strategies 4 and 6, and the higher the coefficient, the lower the corresponding profits will be under the same circumstances.

## Conclusions and limitations

When there are heterogeneous consumers with high price and out-of-stock regret in the market, this paper constructs a model of no-pre-sale, discount-pre-sale and premium-pre-sale, this paper studies the decision-making problem of monopoly retailers with limited capacity under the pre-sale model. The results show that: (1) the regret behavior of consumers does not affect

the price of products when they are not pre-sold, and it does not affect the lower price of products in the two stages of the pre-sale strategy, but it will reduce the higher price when they are pre-sold. The high price regret sensitivity has a negative effect on the higher price of the products in the pre-sale strategy, while the out-of-stock regret has a negative effect on the retailer's profit, though it does not affect the price. (2) when the production capacity is low, retailers should adopt the strategy of high price and no pre-sale to eliminate the negative impact of consumers' regretful behavior, and make use of high unit profits to enhance the overall revenue. (3) when the production capacity is sufficient, the retailer should adopt the limited-quantity discount pre-sale strategy. Among them, if the capacity is relatively low, the proportion of rational consumers is large and the high price regret coefficient is small, the lowest valuation should be sold in advance, the highest valuation should be sold in advance, otherwise the lowest valuation should be sold in advance, a price slightly below the highest valuation is for sale. (4) when the production capacity is sufficient, the retailer should adopt the unlimited premium pre-sale strategy. If the stock-out regret sensitivity coefficient is large, or the stock-out regret sensitivity coefficient is small and the rational consumers account for a large proportion, the highest valuation should be sold in advance, the lowest valuation should be sold now, otherwise it should be sold in advance at a price slightly lower than the highest valuation, the lowest valuation is now for sale.

The above conclusion shows that when the consumers in the market are more affected by regret, retailers should consider the purchase behavior of these consumers to appropriately lower the maximum price to achieve higher returns, otherwise, should ignore this part of consumer demand, high-priced sales. In addition, in order to ensure better profit for retailers, the discount and the limited quantity should be used simultaneously in the pre-sale mode. The limited quantity is to distribute the stock reasonably, and to ensure that the high-valued consumers can buy the products at higher prices at the present sale stage. At the same time, the premium and unlimited quantity should be used at the same time, and can be clearly marked at the time of sale inventory is limited, there will be some consumers even if the purchase will not get the product, this time the unlimited is to encourage more consumers in the pre-sale phase to buy products at high prices.

However, this paper assumes that the consumer valuation follows a two-point distribution, which has some limitations. Future research can set it as a uniform distribution or a random variable, to further explore the impact of consumer regret on the limited capacity of retailers limited pre-sale strategy.

## Supporting information

**S1 Appendix.**
(DOCX)

## Author Contributions

**Writing – original draft:** Yue-yu Qi.

**Writing – review & editing:** Yong-chang Jiang.

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
