## [Decision Letter · Decision Letter 0]

24 Feb 2023

PONE-D-23-01098A study on limited pre-sale strategy with consideration of consumer regretPLOS ONE

Dear Dr. QI,

Thank you for submitting your manuscript to PLOS ONE. After careful consideration, we feel that it has merit but does not fully meet PLOS ONE’s publication criteria as it currently stands. Therefore, we invite you to submit a revised version of the manuscript that addresses the points raised during the review process.

We look forward to receiving your revised manuscript.

Kind regards,

Saad Ahmed Javed, Ph.D

Academic Editor

PLOS ONE

Journal Requirements:

   "Project supported by the Heilongjiang Provincial Social Science Foundation Project(No. 18TQD365)"

Additional Editor Comments:

Major revision required. 

Both reviewers have provided the authors with useful suggestions to improve. Thus, the manuscript needs at minimum a major revision. Please address these comments carefully.

Reviewers' comments:

Reviewer's Responses to Questions

**Comments to the Author**

1. Is the manuscript technically sound, and do the data support the conclusions?

Reviewer #1: Yes

Reviewer #2: Yes

2. Has the statistical analysis been performed appropriately and rigorously? 

Reviewer #1: Yes

Reviewer #2: Yes

3. Have the authors made all data underlying the findings in their manuscript fully available?

Reviewer #1: Yes

Reviewer #2: Yes

4. Is the manuscript presented in an intelligible fashion and written in standard English?

Reviewer #1: No

Reviewer #2: Yes

5. Review Comments to the Author

Reviewer #1: This manuscript constructs a model to study the retailer's optimal limited presale strategy. The topic seems interesting, but there are several major comments as follows:

- The literature review in the introduction section should be extended, and more recent studies should be added.

- The research questions are not clear. The introduction section could be organized much better.

- At the end of the introduction, the manuscript's structure should be explained.

- This manuscript suffers from a poor citation. For example, I suggest the authors check Section 3. How is it possible? Several citations are needed in this section.

- The structure and illustration of the manuscript need significant improvement. I suggest the authors make the manuscript more readable.

- Equation number is essential for all equations. The current format is confusing.

- Where is the proof of Theorem 1? If the authors cannot prove it, they should mention the reference.

Reviewer #2: This paper considers the heterogeneous consumers with regret behavior in the market, and constructs a model to study the retailer's optimal limited pre-salestrategy. My research does not involve this field, but I think the paper is well written after reading it. I have only one suggestion: some important conclusions and theorems need to be explained in the form of diagrams.

6. PLOS authors have the option to publish the peer review history of their article (what does this mean?). If published, this will include your full peer review and any attached files.

Reviewer #1: **Yes: **Amin Mahmoudi

Reviewer #2: No

---

## [Author Response · Author response to Decision Letter 0]

9 Mar 2023

Dear Editor and Reviewers,

Thank you so much for taking the time to review this manuscript. We are very grateful for your comments and suggestions. According to the instructions in your letter, we have uploaded the revised manuscript, and all changes are highlighted by using the revision mode in Word. We enclose our responses to each of the comments made by the academic editor and two commentators. Comments from academic editors and two commentators are reproduced in italics, and our responses are given directly in different color (red) . Thanks Again!

Academic Editor

Q1: Please ensure that your manuscript meets PLOS ONE's style requirements, including those for file naming.

Answer1: Thank you for your suggestions. We have completed two documents according to the template, including the main body of the paper and the introduction of the author, which have been uploaded.

Q2: Thank you for stating the following financial disclosure: "Project supported by the Heilongjiang Provincial Social Science Foundation Project(No. 18TQD365)" .Please state what role the funders took in the study.

Answer2: The funders had no role in study design, data collection and analysis, decision to publish, or preparation of the manuscript.

Q3: In your Data Availability statement, you have not specified where the minimal data set underlying the results described in your manuscript can be found.

Answer3: All the data in the paper are reflected in the manuscript.

Reviewer #1 

Q1: The literature review in the introduction section should be extended, and more recent studies should be added.

Answer1: Thank you for your suggestions, we agree and have updated the content. We isolated the literature review in the introduction as part two. In this paper, the existing research reviews are collated and a new citation is added. Specific modifications are as follows:

 Line 78-106, page 2-3: 

2. Literature review

The literature related to this study mainly deals with two aspects: pre-sale strategy and consumer regret.

2.1. Pre-sale strategy

In the field of pre-sale strategy research, Xie and Shugan's research shows that offer-ing pre-sales is good for businesses, especially when consumers are uncertain about the value of the product [7]. Then Li Hui and Qi Ershi compared the pre-sale and no-pre-sale strategies to give the conditions for retailers to implement the pre-sale strategy [8]. Later scholars will be divided into two kinds of pre-sale strategy research. Among them, Zhao Wen-yan and Ren Xiang-yu studied the premium pre-sale strategy of perishable products based on prospect theory [9], while Duan Yong-rui and others studied the effect of con-sumer regret on premium pre-sale strategy [10], after the Shi Bao-li and other increased returns on the impact of premium pre-sale strategy [11]. In addition, Chen Jing and others studied the retailer's discount pre-sale strategy during the shopping season [12], Yang Xue and others studied the optimal price and order quantity under the deposit pre-sale strate-gy [13], and Xia Yang and others increased the research on the delivery strategy under the deposit pre-sale model [14]. Liu Xu-wang et al studied its effect on the deposit pricing strategy from the perspective of payment pain passivation [15]. However, there is little lit-erature on limited pre-sale. Zhang et al. explored the pre-sale strategy for new products [16], and Yu et al. demonstrated that pre-sale may be advantageous when there is a large number of consumers and a high degree of individual valuation diversification [17]. After that, Ji Guo-jun and so on introduced the reference price effect in the pre-sale model, thus obtained the retailer's optimal pre-sale decision [18], and then continued to study the ref-erence price effect on the limited pre-sale strategy [19]. Sun Cai-hong Sun introduced a two-stage robust newsboy model of limited pre-sale in the environment of uncertain out-put, and discussed the pre-sale strategy under the Shandong stick decision [29].

However, none of the above literature has taken consumer regret as a factor affecting limited pre-sale.

2.2. Consumer regret

In the area of consumer regret research, Özer et al. considered the high price that consumers would face if they bought immediately after the full-price period and the stock-out that they would face if they waited until the liquidation period [20]. Then, Adida et al, on the basis of özer et al, considered the effect of anticipatory regret behavior on firms' pricing strategy choice in the competitive market environment, and emphasized the advantage of price reduction strategy [21]. Chen Feng-jia and Guan Zhen-zhong studied the impact of expected regret on return strategies when consumers' valuations are uncer-tain [22]. Li Qing-qing et al analyzed the impact of high price regret and out-of-stock regret on the pricing strategy of enterprises [23]. In order to explore the optimal pricing of enter-prises, Ye Xinlan and her colleagues constructed a decision-making model under the dy-namic pricing and commitment pricing strategies [24]. Shi Bao-li and other considerations when consumers pre-purchase regret behavior does not provide returns, returns no longer on sale and return re-sale of pre-sale strategy [11]. 

However, these studies did not introduce consumer regret into the study of limited pre-sale.

 Line 101-104, page 3: A new reference has been added

we have added a paragraph to summarize previous studies and linked my research questions to the existing research conversation as: 

Sun Cai-hong Sun introduced a two-stage robust newsboy model of limited pre-sale in the environment of uncertain out-put, and discussed the pre-sale strategy under the Shandong stick decision [29].

Newly added literatures are:

SUN Caihong, LI Zhen, YU Hui4. Robust Strategy Analysis of Capacity Rationing in Advance Selling About Perishable Products Under Yield Uncertainty [J/OL]. Journal of Systems Science and Mathematical Scienc-es:1-15[2023-03-09].http://kns.cnki.net/kcms/detail/11.2019.O1.20230112. 0959.008.html

Q2: The research questions are not clear. The introduction section could be organized much better.

Answer2: Thank you for your suggestions, we agree and have updated the content. We have revised “introduction”. Specific modifications are as follows:

 Line 24-37, page 1: The first paragraph has been revised and 4 citations have been added.

Pre-sale mode is a kind of sales mode in which the seller allows the buyer to buy the goods or services at a certain time before they are consumed, and the buyer has to wait for a certain period time before he can consume the goods or services. For consumers, buying in advance not only allows them to get discounts or lock-in goods ahead of time to avoid the risk of out-of-stock but also because retailers often promise to pay first and deliver first, choosing to pre-order can get goods or services earlier. For retailers, the advance sale method can help them better forecast the demand for goods [1], so that they can stock up in the spot period to effectively reduce the inventory risk [2]: when the consumer demand is uncertain and the preference is not clear, pre-sale data can mitigate Information asym-metry risks between retailers and consumers [3,4] , and can be used to provide volume and profit through price discrimination [5]. With the development and popularization of online shopping, advance sale is more and more popular with retailers. Therefore, when to adopt what kind of pre-sale strategy and how to price has become a limited capacity of retailers to solve the problem.

Newly added literatures are:

Mao Z F，Dong Z C，Liu W，et al． Comparative research on joint strategy of advance selling and buy-back to con-ventional products and new products［J］．Computer integrated manufacturing systems,2017,23(04) : 867-873．

Moe W W，Fader P S．Fast-track: Article using advance purchase orders to forecast new product sales［J］． Marketing science,2002,21(3) : 347-364．

Nocke V，Peitz M，Rosar F． Advance-purchase discounts as a price discrimination device［J］．Journal of Economic Theory,2011,146(1) : 141-162．

Shugan S M，Xie J H． Advance selling for service［J］． California Management Review,2004,46(3) : 37-54．

 Line 38-64, page 1-2: 

Pre-sale strategy can be divided into discount pre-sale and premium pre-sale ac-cording to the relationship between the pre-sale period and the spot price. Premium pre-sale refers to the pre-sale when the price of the product is higher than the current price. Because for a limited number of service products, in order to avoid the risk of spot short-age, consumers are often willing to pay a premium over the spot price in advance to lock-in products. For example, loyal users of some electronic products are willing to pay a premium for access to the product in the first place. On the contrary, retailers will lower prices during the pre-sale stage to encourage consumers to buy products during the pre-sale stage. For the study object of this paper, that is, the limited capacity or service capacity and can not be improved in a short period of time, under certain conditions to limit the pre-sale period of sales can make retailers increase revenue [6], that is, limited pre-sale. Under the constraint of production capacity, restricting the pre-sale volume has become a popular operation mode under the pre-sale mode.

The intertemporal price discrimination of retailers may increase the income, but it will affect the consumer's purchasing decision. Therefore, intertemporal price combination should consider the consumer's behavior. From the perspective of consumers, on the one hand, the choice of pre-sale purchase can ensure the availability of products, but once the normal sale phase can buy goods, consumers will pay more money for pre-sale period regret; On the other hand, if consumers choose to wait until the normal sale stage to buy, then will face the possibility of out-of-stock, once out of stock, consumers will regret their waiting behavior, bemoan the loss of utility caused by waiting. Because of the change in utility caused by regret, strategic consumers begin by calculating the expected utility of their purchases at various stages in order to choose to buy at the time of their Utility maximization problem.

In this paper, we will study the limited pre-sale strategy of retailers with limited capacity or limited service capacity, and focus on the optimal decision-making under different capacity conditions.

Q3: At the end of the introduction, the manuscript's structure should be explained.

Answer3: Thank you for your suggestions, we agree and have updated the content. We added the manuscript's structure at the end of the introduction:

Line 71-77, page 2: 

The rest of this article is organized as follows. The second part is literature review. The third part describes the specific parameters and assumptions of the model. In the fourth part, the profit functions of non-pre-sale, premium pre-sale and discount pre-sale are constructed. The fifth part compares the advantages and disadvantages of the three strategies, and obtains the retailer's strategies under different conditions. The sixth part summarizes the research content of this paper, and puts forward the deficiency and the research direction in the future.

Q4: This manuscript suffers from a poor citation. For example, I suggest the authors check Section 3. How is it possible? Several citations are needed in this section.

Answer4: Thank you for your suggestions, we agree and have updated the content. In section 4, Model Construction and Analysis, four more references are added:

 Line 175-177, page 4: 

If the retailer adopts the no-pre-sale strategy, the price is equal and the quantity is unlimited [19], so the consumers have no regret loss, then the purchasing utility of both kinds of consumers is equal U=v-P [27]. 

 Line 200, page 4: 

this is when the retailer gets the most profit [19].

 Line 297, page 6: 

……that the consumer can get the product at the present sale stage [19].

Q5: The structure and illustration of the manuscript need significant improvement. I suggest the authors make the manuscript more readable.

Answer5: Thank you for your suggestions, we agree and have updated the content. We added a literature review section and modified the table header of the table. Specific modifications are as follows:

 Line 78-122, page 2-3: Same as the Answer1:

 Line 412, page 9: Delete C and change “profit” to “C”

 Line 438, page 10: Delete C and change “profit” to “C”

Q6: Equation number is essential for all equations. The current format is confusing.

Answer6: Thank you for your suggestions, we agree and have updated the content. In combination with the comments of another reviewer, we present the profit function formulas for all strategies in Table 1. Specific modifications are as follows:

Line 388-391, page 8-9:

4.4 The profit function of three pre-sale strategies

Combined with Theorem 1-6, the profit functions of three pre-sale strategies shown in Table 1 are obtained.

Table 1. the profit functions of three pre-sale strategies

Q7: Where is the proof of Theorem 1? If the authors cannot prove it, they should mention the reference.

Answer7: Thank you for your suggestions, we agree and have updated the content. We add a proof of theorem 1 in the Appendix A. Specific modifications are as follows:

 Line 500-504, page 12: 

Appendix A.1 Proof of Theorem 1.

Compare the profit function for strategy 1 and Strategy 2:

When C≤1/2, 〖π_1〗^*-〖π_2〗^*=V_L C-1/2 V_H<0, Strategy 2 is better;

When 1/2<C≤V_H/(2V_L ), 〖π_1〗^*-〖π_2〗^*=(V_L-V_H )C<0, Strategy 2 is better;

When C>V_H/(2V_L ), 〖π_1〗^*-〖π_2〗^*=V_L C-1/2 V_H>0, Strategy 1 is better;

Reviewer #2 

Q: some important conclusions and theorems need to be explained in the form of diagrams.

Answer: Thank you for your suggestions, we agree and have updated the content. We present the profit function formulas for all strategies in Table 1. Specific modifications are as follows:

Line 388-391, page 8-9:

4.4 The profit function of three pre-sale strategies

Combined with Theorem 1-6, the profit functions of three pre-sale strategies shown in Table 1 are obtained.

Table 1. the profit functions of three pre-sale strategies

---

## [Decision Letter · Decision Letter 1]

3 Apr 2023

PONE-D-23-01098R1考虑消费者后悔的有限预售策略研究PLOS ONE

Dear Dr. QI,

Thank you for submitting your manuscript to PLOS ONE. After careful consideration, we feel that it has merit but does not fully meet PLOS ONE’s publication criteria as it currently stands. Therefore, we invite you to submit a revised version of the manuscript that addresses the points raised during the review process.

We look forward to receiving your revised manuscript.

Kind regards,

Chaohai Shen

Academic Editor

PLOS ONE

Journal Requirements:

Additional Editor Comments (if provided):

Dear Authors,

The revised version is much better than the original one. You have made possible changes following reviewers' comments. However, there are still three minor parts I hope you can adjust before I consider the possibility of accepting the manuscript.

First, you need to properly adjust the literature source. This paper is not about any specific thing only observed in China but a topic commonly explored by researchers across the world. It’s fine to cite some papers written in languages other than English. However, in your case, there are numerous papers published in prestigious international journals discussing relevant topics, while the Chinese papers you cited are not essential references in the relevant fields. It is usually hard to convince readers that your findings are academically important without solid literature support. Accordingly, you must further revise the literature review section.

In addition, please carefully read papers published in relevant prestigious international journals to make sure that you follow the common standards to cite other researchers' work in your manuscript.

Second, you must correct the format of the reference and possibly other relevant parts. I can see that you have made some changes, but there are still concerns. Usually, you can download the latest relevant papers published in PLOS ONE to check the format. For example, why you put "[J]" aside the title of a paper? Also, why the format for authors' names in the reference part are so inconsistent? There are more problems in the format of your manuscript and you must carefully correct all of them.

Finally, I'm not sure the reason but the title shown in your submission is in Chinese. You may check with the Journal's staff to make sure it's acceptable.

Please try to submit the possibly best revised version to me when you are ready.

Sincerely,

Reviewers' comments:

Reviewer's Responses to Questions

**Comments to the Author**

1. If the authors have adequately addressed your comments raised in a previous round of review and you feel that this manuscript is now acceptable for publication, you may indicate that here to bypass the “Comments to the Author” section, enter your conflict of interest statement in the “Confidential to Editor” section, and submit your "Accept" recommendation.

Reviewer #1: All comments have been addressed

Reviewer #2: All comments have been addressed

2. Is the manuscript technically sound, and do the data support the conclusions?

Reviewer #1: No

Reviewer #2: Yes

3. Has the statistical analysis been performed appropriately and rigorously? 

Reviewer #1: Yes

Reviewer #2: Yes

4. Have the authors made all data underlying the findings in their manuscript fully available?

Reviewer #1: Yes

Reviewer #2: Yes

5. Is the manuscript presented in an intelligible fashion and written in standard English?

Reviewer #1: Yes

Reviewer #2: Yes

6. Review Comments to the Author

Reviewer #1: The authors addressed my comments.There are no more comments from my side. It can be accepted in its current form.

Reviewer #2: Tha authors address all my concerns, so the currently manuscript can be accepted for publication in the Plos One.

7. PLOS authors have the option to publish the peer review history of their article (what does this mean?). If published, this will include your full peer review and any attached files.

Reviewer #1: No

Reviewer #2: No

---

## [Author Response · Author response to Decision Letter 1]

6 Apr 2023

Dear Editor and Reviewers,

Thank you so much for taking the time to review this manuscript. We appreciate your comments and suggestions. According to the instructions in your letter, we have uploaded the revisions, all of which are highlighted using the change mode in Word. We enclose our responses to the comments made by the academic editor and two commentators. Comments from academic editors are reproduced in italics, and our responses are given directly in a different color (red) . Thanks Again!

Academic Editor

Q1: First, you need to properly adjust the literature source.

Answer1: Thanks for your suggestion, we agreed and updated the content. We rearranged the existing research reviews and added new citations. Amend it as follows:

(1) Line 91-135, page 3-4: 

Early research on pre-sale models focused on the service industry, commonly pre-sales of various types of tickets. Jinhong X and Steven M showed that offering pre-sales is beneficial for firms, especially when consumers are uncertain about the value of the product [7]. Yu M et al. demonstrated that pre-sale may be advantageous when there is a large number of consumers and a high degree of individual valuation diversification [8]. Li C et al explored the pre-sale decision in a pre-sale model when both short-sighted and strategic consumers are present [9]. Then Li H and Qi ES comparatively studied pre-selling and non-pre-selling strategies so as to give the conditions for retailers to implement pre-selling strategies [10]. However, there is little literature on limited pre-sale. After that, Chen J and Bell PC studied the demand uncertainty of fixed capacity and limited product choice for fugitive products [11]. Zhang M et al explored the limited pre-sale model for new products on flash sale platforms [12].Ji GJ and so on introduced the reference price effect in the pre-sale model, thus obtained the retailer's optimal pre-sale decision [13], and then continued to study the ref-erence price effect on the limited pre-sale strategy [14]. 

(2) Line 137-153, page 4:

Expected regret reflects a combination of regret, depression, disappointment, remorse, and self-blame that will be experienced by not buying the limited product [15,16,17], coupled with the higher perceived value of the product that comes with a production-based limited [18] and the stronger motivation for consumers to own the product [19]. Thus, compared to sales-based limits, production-based limits are more likely to induce strong feelings of expected elation and expected regret. In the area of consumer regret research, Zen Z and Zheng Y considered the high price that consumers would face if they bought immediately after the full-price period and the stock-out that they would face if they waited until the liquidation period [20]. Then, Adida E and ÖZER Ö, on the basis of Zen Z and Zheng Y, considered the effect of anticipatory regret behavior on firms' pricing strategy choice in the competitive market environment, and emphasized the advantage of price reduction strategy [21]. Chen FJ and Guan ZZ studied the impact of expected regret on return strategies when consumers' valuations are uncertain [22]. Zhou JH et al analyzed the impact of high price regret and out-of-stock regret on the pricing strategy of enterprises [23]. Shi BL and other considerations when consumers pre-purchase regret behavior does not provide returns, returns no longer on sale and return re-sale of pre-sale strategy [24]. 

Q2: Second, you must correct the format of the reference and possibly other relevant parts.

Answer2: Thank you for your suggestions, we agree and have updated the content. We have revised “introduction”. Specific modifications are as follows:

(1) Line 632-906, page 18-19: The first paragraph has been revised and 4 citations have been added.

1. Song JS, Paul H, Zipkin. Newsvendor problems with sequentially revealed demand information. Naval Research Logistics. 2012:59(8): 601-612.

2. Christopher S, Tang, Kumar R, Aydin A, Jihong O. The Benefits of Advance Booking Discount Programs: Model and Analysis. Management Science. 2004;50(4) : 465-478. 

3. Mao ZF, Dong ZC, Liu W. Comparative research on joint strategy of advance selling and buy-back to conventional products and new products. Computer integrated manufacturing systems. 2017;23(04) : 867-873.

4. Wendy W. Moe,Peter S. Fader. Fast-track: Article using advance purchase orders to forecast new product sales.Marketing science.2002;21(3) : 347-364.

5. Volker N,Martin P,Frank R.Advance-purchase discounts as a price discrimination device. Journal of Economic Theory.2011;146(1) : 141-162.

6. Shugan SM,Xie J. Advance selling for service.California Management Review.2004;46(3) : 37-5.

7. Jinhong X and Steven M. Shugan. Electronic Tickets, Smart Cards, and Online Prepayments: When and How to Advance Sell. Marketing Science. 2001;20(3) : 219-243.

8. Yu M, Kapuscinski R, Ahn HS. Advance Selling: Effects of Interdependent Consumer Valuations and Seller’s Capacity. Management Science. 2015; 61(9):2100-2117.

9. Li C, Zhang F. Advance demand information, price discrimination,and preorder strategies. Manufacturing & Service Operations Management.2013;15(1) : 57-71.

10. Li H, Qi ES. Advance Sellingin the Presence of Uncertainty Market Size. Chinese Journal of Management Science.2017;25(02):50-56. 

11. Chen J, Bell PC. Enhancing revenue by offering a flexible product option. International Transactions in Operational Research. 2017;24(4): 801–820.

12. Zhang M,Cheng T,Du J. Advance selling of new products to strategic consumers on flash sale platforms. International Journal of Logistics: Research and Applications.2018;21(3) : 318-331.

13. Ji GJ, Sun ZF. Advance selling with reference price effect and consumer heterogeneity. Systems Engineering-Theory & Practice. 2018; 38(12): 3059-3070.

14. Ji GJ, Sun ZF ,Yang GY, Zeng QY, et al. Advance selling with consumer reference price effect and limited advance sales. Journal of Industrial Engineering and Engineering Management. 2021;35(04): 178-189.

15. Bagozzi,Utpal M. Dholakia,Suman B. How effortful decisions get enacted: The motivating role of decision processes, desires, and anticipated emotions. Journal of Behavioral Decision Making. 2003;16(4):273－295. 

16. Perugini M,Bagozzi RP. The role of desires and anticipated emotions in goal-directed behaviours: Broadening and deepening the theory of planned behavior. The British Journal of Social Psychology.2001;40(1):79－98.

17. Proksch M, Orth UR, Cornwell TB. Competence enhancement and anticipated emotion as motivational drivers of brand attachment. Psychology & Marketing. 2015;32(9) : 934－949.

18. Jang WE, Ko YJ, Morris JD,et al. Scarcity message effects on consumption behavior: Limited edition product considerations. Psychology & Marketing.2015;32(10):989－1001.

19. Tian KT, Bearden WO, Hunter GL. Consumers’need for uniqueness: Scale development and validation. Journal of Consumer Research. 2001; 28(1):50－66.

20. Zen Z, Zheng Y. Markdown or Everyday Low Price? The Role of Behavioral Motives. Management Science. 2016; 62(2): 326-346.

21. Adida E, ÖZER Ö. Why Markdown as a Pricing Modality?. Management Science. 2019; 65(5):2161-2178.

22. Chen SJ, Guan ZZ. Return policy for online retailers in the presence of product value uncertainty and consumer anticipated regret. Journal of Industrial Engineering and Engineering Management. 2022; 36(05): 181-195.

23. Zhou JH, Li QQ, Xu XL. Pricing Strategy of Fashion Products with Strategic Consumer Regret. Chinese Journal of Management Science. 2022;30(11):117-126.

24. Shi BL, Xu Q, Sun ZM. Pre-sale and Return Strategy of Retailers Considering Consumer Regret Behavior. Operations Research and Management Science. 2022; 31(10): 40-46.

25. Wang XH, Zeng C. A model of advance selling with consumer heterogeneity and limited capacity. Journal of Economics.2016;117(2):137-165. 

26. Xu J, Duan Y. Pricing, ordering, and quick response for online sellers in the presence of consumer disappointment aversion. Transportation Research Part E: Logistics and Transportation Review. 2020;137: 101925.

Q3: Finally, I'm not sure the reason but the title shown in your submission is in Chinese. You may check with the Journal's staff to make sure it's acceptable.

Answer3: Thank you for your suggestions, we agree and have updated the content. We will correct the title when we submit this revision.

---

## [Editor Report · Decision Letter 2]

14 Apr 2023

A study on limited pre-sale strategy with consideration of consumer regret

PONE-D-23-01098R2

Dear Dr. QI,

We’re pleased to inform you that your manuscript has been judged scientifically suitable for publication and will be formally accepted for publication once it meets all outstanding technical requirements.

Kind regards,

Chaohai Shen

Academic Editor

PLOS ONE
---

## [Editor Report · Acceptance letter]

24 Apr 2023

PONE-D-23-01098R2 

A study on limited pre-sale strategy with consideration of consumer regret 

Dear Dr. QI:

I'm pleased to inform you that your manuscript has been deemed suitable for publication in PLOS ONE. Congratulations! Your manuscript is now with our production department. 

Kind regards, 

on behalf of

Dr. Chaohai Shen 

Academic Editor

PLOS ONE